# Do electronic medication monitors improve tuberculosis treatment outcomes? Programmatic experience from China

**Ni Wang** [1], **Hemant Deepak Shewade** [2,3], **Pruthu Thekkur** [2,3], **Hui Zhang** [1], **Yanli Yuan** [4], **Xiaomeng Wang** [5], **Xiaolin Wang** [6], **Miaomiao Sun** [7], **Fei Huang** [1]*

1 National Center for Tuberculosis Control and Prevention, Chinese Center for Disease Control and Prevention, Beijing, China, 2 International Union against Tuberculosis and Lung Disease (The Union), Paris, France, 3 The Union South East Asia Office, New Delhi, India, 4 Jilin Research Institute of Tuberculosis Control, Changchun, China, 5 Zhejiang province Center Disease Control and Prevention, Hangzhou, China, 6 The Fourth People's Hospital of Ningxia Hui Autonomous Region, Yinchuan, China, 7 PATH China Program, Beijing, China

* huangfei@chinacdc.cn

**Data Availability Statement:** All relevant data are within the manuscript and its Supporting information files.

## Abstract

### Background

In China, an indigenously developed electronic medication monitor (EMM) was used. EMM recorded each time the device was opened (no real time data), offering an indirect measure of tuberculosis treatment adherence. Previous study in China showed that the EMM uptake was satisfactory, missing adherence data were common in the information management system (25%) and shift to directly observed therapy (DOT) based on poor adherence documented by EMMs were seldom.

### Objectives

Among people with tuberculosis notified in 30 counties (July-December 2018) where EMM supported self-administered therapy (SAT) was suggested to all eligible (no communication impairment, ambulatory), we assessed the relative differences in unfavourable outcomes and deaths among those started on EMM at baseline (within first month of diagnosis) when compared to SAT alone.

### Methods

This was a cohort study using secondary data. We employed an intention to treat analysis, and used modified Poisson regression with robust variance estimates to assess the association.

### Results

Of 1810 eligible people, 1047 used EMM at baseline and of them, 216 (20.1%) stopped using EMM midway. Of 763 people who did not use EMM at baseline, 267 (35.0%) started using EMM later during the treatment. Among those who started using EMM at baseline,

**Funding:** The SORT IT training programme under which this manuscript was developed was funded by the Department for International Development (DFID), UK. The publications costs for this manuscript were supported by the China-Gates Foundation TB Project (OPP1137180). The funders had no role in study design, data collection and analysis, decision to publish, or preparation of the manuscript.

**Competing interests:** The authors have declared that no competing interests exist.

6.3% [95% CI: 4.9, 8.0] had unfavourable outcomes compared to 6.7% [95% CI: 5.1, 8.8] among those who did not (p = 0.746). Lesser deaths were observed in people who started EMM at baseline when compared to those who did not: 2.5% [95% CI: 1.7, 3.7] versus 3.5% [95% CI: 2.4, 5.2], p = 0.191. The lack of association remained after adjusting for potential confounders (occupation, TB classification and TB category).

## Conclusion

Under programmatic settings, we did not find significant differences in the outcomes. Optimization of EMMs by shifting to DOT when indicated, addressing the issue of missing data and ensuring continuous use is required.

## Introduction

Globally, 85% of people with new and relapse tuberculosis (TB) successfully completed treatment while it was 61% among previously treated people (2017 cohort) [1]. Adherence to TB treatment is essential to attain treatment success. To ensure treatment adherence, World Health Organization recommends community-based or home-based directly observed treatment (DOT) over health facility based DOT or self-administered therapy (SAT). DOT administered by trained lay providers or health-care workers is recommended over DOT administered by family members or SAT [2].

SAT has lower treatment success rate when compared to DOT alone, but with no significant difference in mortality [3]. TB treatment outcomes improved with the use of adherence interventions when compared to DOT or SAT alone [3]. Hence, if used, SAT has to be implemented with certain adherence support mechanisms. The adherence support includes one or more of the following: patient education, staff education, material support, psychological support, incentives, reminders, tracers and digital support technologies [2].

Three digital technologies are recommended by World Health Organization as adherence support: short message service, video observed treatment and electronic medication monitors (EMM) [4]. There is a very modest effect of short message service on outcomes and the quality of evidence is low [5–7]. In India, the strategy of the patient giving a missed call after every dose intake showed suboptimal accuracy for monitoring treatment compliance and also was not associated with improved outcomes. Non-receipt of missed call was not followed up with appropriate action by the healthcare providers [8]. Additionally, video observed treatment had no evidence of a difference in rates of treatment completion and mortality when compared to DOT [9, 10].

EMMs are automated electronic devices that monitor and store adherence records combined with audible reminder alarms [11]. The EMMs can function with or without mobile broadband internet coverage and the use of EMMs reduces health workers to engage with TB patients on daily basis for DOT. Thus, the EMMs are looked as accessible and affordable solutions for adherence monitoring in resource-limited settings [4]. In rural Morocco, medication event monitoring system (real time data shared by monitoring the weight of drugs in the medication monitor) with SAT increased treatment success and decreased the loss to follow-up when compared to SAT alone [12].

Universal implementation of DOT in a country like China is impossible. A systematic review in China found that only 50% of people with TB received DOT (from a health care worker or a family member) and 50% were on SAT without any adherence support [13].

Hence, the Chinese Center for Disease Control and Prevention (China CDC) explored the use of an indigenously developed EMM to support SAT from 2009 [14]. These EMMs are automated electronic devices without mobile broadband internet coverage that record (within the device) each time the device is opened, offering an indirect measure of retrospective drug consumption assessment to the health care provider during the hospital visits.

Under trial settings in China, these EMMs reduced poor medication adherence by 40–50% compared to the standard of care in China's National Tuberculosis Control Program [6]. Previous studies have also shown a high level of acceptability and satisfaction among the people with TB and health care workers [14, 15]. Under programme settings, EMM uptake was satisfactory, but instances of missing EMM data in the information management system (EMM-IMS) were common (25%). Furthermore, nearly in four-fifth instances, people were not shifted to DOT although objective evidence of non-adherence was available [16].

However, the evidence on the impact of EMMs on TB treatment outcomes is still limited [17], especially of those EMMs that do not provide real-time data. In this paper we assessed the relative differences in unfavourable TB treatment outcomes and deaths among those started on EMM when compared to SAT alone.

## Methods

### Study setting

China has the world's second highest tuberculosis burden, with an estimated 866,000 cases in 2018 [1]. The treatment success rate among people with new and relapse TB is 93% and among people with previously treated TB is 83% (2017 cohort) [1].

A total of 30 counties were involved in this study, which come from three provinces: nine from Zhejiang province (eastern region), sixteen from Jilin province (middle region) and five from Ningxia Autonomous Region (west region). The economic development is higher in eastern regions and lower in western region. The prevalence of TB, by contrast, is higher in western region and lower in eastern region.

### Study design and population

This was a cohort study using secondary data (routinely collected in national TB Programme) in 30 pilot counties of China. We used an intention to treat design. We classified eligible people into two groups based on EMM use at baseline (within first month of diagnosis). Those who did not use EMM at baseline included those who later started using EMM after first month or never used EMM. Those who used EMM at baseline also included those who stopped using EMM midway. We decided against stratifying based on 'ever EMM use' because of the possibility of survival bias among people who started using EMM later during the treatment. This could possibly result in a biased association of 'ever EMM use' with more favourable outcomes. For the same reason, we did not classify the people based on 'duration of using EMM'.

People having no communication impairment (mental, visual, auditory, or speech) and not requiring admission at notification were identified as 'eligible' to use EMM. All the 'eligible' people with TB routinely notified in web-based TB information management system (TBIMS) between July and December, 2018 were included and suggested EMM supported SAT by the doctors from TB designated hospitals (county level, TB basic management unit starting at county level). We excluded people known to be rifampicin-resistant or multidrug-resistant, and people transferred into a county.

The EMM was designed to monitor treatment adherence throughout a one-month fixed-dose regimen. The structure of the EMM and quality control protocol has been described elsewhere [16].

## Management of TB with and without EMM

All the people with TB received daily fixed-dose combination treatment over six to 12 months (see Table 1). People with TB would mark daily intake of medication in their treatment cards. The village doctors (village level licensed general practitioners) would visit every ten days during the intensive phase followed by once a month to check on their health status and treatment adherence. People on treatment visited the TB designated hospital on a monthly basis where the doctor would assess adherence based on their treatment card. If <20% of doses were missed, the person was counselled on importance of treatment adherence. If 20–49% of doses were missed, the frequency of home visits by village doctors was increased to once every seven days for rest of the treatment. If there was continued instance of missing 20–49% of doses or a single instance of missing ≥50% of doses, people were shifted to DOT administered by village doctors [14].

For EMM eligible people, the verbal consent of using EMM was obtained at the TB designated hospitals. The trained doctors programmed the EMMs before starting on outpatient treatment. During the monthly visits to the TB designated hospital, doctors generated the adherence report by connecting the EMM to offline software in their computers and took appropriate action as described before. The data from the offline system were uploaded to the EMM-IMS by the TB designated hospital staff.

## Data variables and sources of data

We extracted data variables from TBIMS (diagnosis date, treatment start date, sex, age, occupation, migrant status, category of TB, classification of TB, treatment outcomes), EMM-IMS (date of starting EMM) and paper-based patient records at county (to assess edibility for EMM). We derived the following variables: treatment initiation delay (in days) and EMM use at baseline (yes/no). We classified cure and treatment completion as favourable treatment outcomes and loss to follow up, death, treatment failure, transfer to drug-resistant TB treatment and not evaluated as unfavourable outcomes (see Table 2) [18].

## Data analysis

We used STATA (version 12.1, copyright 1985–2011 Stata Corp LP USA) for analysis. Separate intention to treat analysis was done for unfavourable outcomes and death (outcomes). We summarized them as proportions with 95% confidence intervals (CI). We identified potential confounders for the association between EMM use at baseline (yes–exposure of interest) and the outcomes using the following criteria: associated with the exposure of interest (p<0.2) and associated with the outcome (unfavourable outcome / death—p<0.20). Age and sex were

**Table 1. Treatment regimens for drug-susceptible TB used in the national TB programme, China (2018–19).**

| Type | Regimen |
|---|---|
| New TB | 2HRZE/4HR |
| Previously treated TB | 2HRZES/6HRE or 3HRZE/6HRE |
| Pleurisy | 2HRZE/7HRE or 2HRZE/10HRE |

TB—tuberculosis; H—isoniazid; R—rifampicin; Z—pyrazinamide; E—ethambutol; S–streptomycin.

**Table 2. Operational definition of TB treatment outcomes used in the study, China (2018).**

| Outcome | Definition |
|---------|-----------|
| Cured | A pulmonary TB patient with bacteriologically confirmed TB at the beginning of treatment who was smear- or culture-negative in the last month of treatment and on at least one previous occasion. |
| Treatment completed | People with TB who completed treatment without evidence of failure but with no record to show that sputum smear or culture results in the last month of treatment and on at least one previous occasion were negative, either because tests were not done or because results are unavailable. |
| Treatment failed | People with TB whose sputum smear or culture is positive at month 5 or later during treatment. |
| Died | People with TB who dies for any reason before starting or during the course of treatment. |
| Lost to follow-up | People with TB who did not start treatment or whose treatment was interrupted for 2 consecutive months or more. |
| Not evaluated | People with TB for whom no treatment outcome is assigned. This includes people "transferred out" to another treatment unit as well as people for whom the treatment outcome is unknown to the reporting unit. |
| Transferred to DR-TB* | People notified as drug susceptible TB and then drug-resistant TB is detected during treatment |
| Treatment success | The sum of cured and treatment completed. |
| Unfavourable outcome | All outcomes other than cured and treatment completed |

TB—tuberculosis; DR-TB—drug resistant TB.

* If transferred to DR-TB care during TB treatment and there is evidence of registering in the prefecture level DR-TB center, the person will be excluded from this cohort. If there is no evidence of registering in the prefecture level DR-TB center, the person will be included in this cohort and reported under unfavorable outcomes.

included irrespective of their p values as they are universal confounders. Association between exposure of interest and outcome was assessed using crude and adjusted relative risks (95% CI). Adjusted analysis was done using modified Poisson regression with robust variance estimates.

## Ethics

The ethics committees of the China CDC (number 201909 dated 18 April 2019) and the Ethics Advisory Group of the International Union Against Tuberculosis and Lung Disease (The Union), Paris, France (EAG no 15/19 dated 01 April 2019) approved the study. As the study involved use of secondary programme data, we sought waiver for written informed consent and this was approved by the ethics committees.

## Results

There were 2294 people notified with TB and 2227 (97.1%) had EMM eligibility related data. Of 2227 people, 417 (18.7%) were not eligible for using EMM.

Of 1810 'EMM eligible' people, 1047 used EMM at baseline and of them, 216 (20.1%) stopped using EMM midway. Of 763 people who did not use EMM at baseline, 267 (35.0%) started using EMM later during the treatment.

The baseline characteristics of 1810 people, stratified by EMM use from baseline, are depicted in Table 3. The characteristics were not significantly different but for occupation, TB classification and category. These three variables were also associated with unfavourable

**Table 3. Socio-demographic, clinical and treatment accessibility related characteristics of notified people with TB in the 30 selected counties of China between July and December, 2018, stratified by EMM use at baseline***.

| Characteristics | EMM use at baseline* | | | | p value@ |
|---|---|---|---|---|---|
| | Yes | | No | | |
| | N | (%) | N | (%) | |
| **Total** | **1047** | **(100)** | **763** | **(100)** | |
| Age in years | | | | | |
| <15 | 3 | (0.3) | 5 | (0.7) | 0.131 |
| 15–44 | 377 | (36.0) | 252 | (33.0) | |
| 45–64 | 393 | (37.5) | 274 | (35.9) | |
| > = 65 | 274 | (26.2) | 232 | (30.4) | |
| Sex | | | | | |
| Male | 716 | (68.4) | 505 | (66.2) | 0.335 |
| Female | 331 | (31.6) | 258 | (33.8) | |
| Occupation | | | | | |
| Farmers and herdsmen | 507 | (48.4) | 309 | (40.5) | 0.006** |
| Semi-skilled employee | 97 | (9.3) | 71 | (9.3) | |
| Salary employee | 101 | (9.6) | 93 | (12.2) | |
| Unemployed | 233 | (22.3) | 217 | (28.4) | |
| Studying | 62 | (5.9) | 38 | (5.0) | |
| Others | 47 | (4.5) | 35 | (4.6) | |
| Migrant^ | | | | | |
| No | 997 | (95.2) | 736 | (96.5) | 0.238 |
| Yes | 50 | (4.8) | 27 | (3.5) | |
| Classification | | | | | |
| Bacteriologically confirmed PTB | 584 | (55.8) | 396 | (51.9) | 0.001** |
| Clinically diagnosed PTB | 394 | (37.6) | 278 | (36.4) | |
| Pleurisy | 69 | (6.6) | 89 | (11.7) | |
| Category | | | | | |
| New | 1008 | (96.3) | 708 | (92.8) | 0.001** |
| Previously treated | 39 | (3.7) | 55 | (7.2) | |
| Time interval from diagnosis to treatment (in days) | | | | | |
| Zero | 1002 | (95.7) | 729 | (95.5) | 0.907 |
| ≥ One | 45 | (4.3) | 34 | (4.5) | |

Column percentages.

TB—tuberculosis; PTB—pulmonary TB; EMM- electronic medication monitor.

*Of 1810 notified people with TB who were eligible to use EMM, 1047 used EMM within first month of diagnosis (baseline). The remaining 763 who did not use EMM at baseline.

^ Migrant defined as person staying in the same prefecture for less than six months.

@ Chi squared test.

**p<0.05.

outcomes as well as death (p<0.20, data not shown) and were therefore considered potential confounders (along with age and sex–potential confounders).

The treatment outcomes, stratified by EMM use at baseline, are depicted in Table 4. Among those using EMM at baseline, 6.3% [95% CI: 4.9, 8.0] had unfavourable outcomes compared to 6.7% [95% CI: 5.1, 8.8] among those not using EMM at baseline (p = 0.746). Lesser deaths were observed in people who started EMM at baseline when compared to those who did not:

**Table 4. Treatment outcomes of notified people with TB in the 30 selected counties of China between July and December, 2018, stratified by EMM use at baseline\*.**

| Treatment outcomes | EMM use at baseline\* | | | |
|---|---|---|---|---|
| | Yes | | No | |
| | n | (%) | n | (%) |
| **Total** | **1047** | **(100.0)** | **763** | **(100.0)** |
| Favourable | | | | |
| Cured | 493 | (47.1) | 346 | (45.3) |
| Treatment completed | 488 | (46.6) | 366 | (48.0) |
| Unfavourable | | | | |
| Treatment Failure | 10 | (1.0) | 3 | (0.4) |
| Died | 26 | (2.5) | 27 | (3.5) |
| Not evaluated | 30 | (2.8) | 21 | (2.8) |

Column percentages.

TB—tuberculosis; EMM- Electronic Medication Monitor.

\*Of 1810 notified people with TB who were eligible to use EMM, 1047 used EMM within first month of diagnosis (baseline). The remaining 763 who did not use EMM at baseline.

2.5% [95% CI: 1.7, 3.7] versus 3.5% [95% CI: 2.4, 5.2], p = 0.191. The lack of association remained after adjusting for potential confounders (see Table 5).

## Discussion

Globally, this is the first study assessing the association between TB treatment outcomes and the use of an EMM that does not provide real-time data under programmatic conditions. TB treatment outcomes did not change overall, but there were 33% reductions in deaths, though not statistically significant.

 Our findings are consistent with the findings in trial setting in China using the same EMM. However, the primary study endpoint of the trial was treatment adherence, the sample size was not sufficiently powered to look for differences in treatment outcomes [6]. Meanwhile, our findings are in contrast to the study from Morocco and South African. In rural Morocco, a medication event monitoring system with SAT increased treatment success and decreased the loss to follow-up among people with new smear positive pulmonary TB when compared to

**Table 5. Association between EMM use at baseline\* and TB treatment outcomes in the 30 selected counties of China between July and December, 2018.**

| Treatment outcomes | EMM use at baseline\* | | RR^(0.95 CI) | aRR^ (0.95 CI)\*\* |
|---|---|---|---|---|
| | Yes | No | | |
| | % (outcome/total) | % (outcome/total) | | |
| Unfavourable | 6.3 (66/1047) | 6.7 (51/763) | 1.00 (0.98, 1.03) | 1.00 (0.98, 1.03) |
| Death | 2.5 (26/1047) | 3.5 (27/763) | 0.70 (0.41, 1.19) | 0.74 (0.44, 1.26) |

TB—tuberculosis; EMM—Electronic Medication Monitor; RR—relative risk; aRR—adjusted RR; CI—confidence interval.

\*Of 1810 notified people with TB who were eligible to use EMM, 1047 used EMM within first month of diagnosis (baseline). The remaining 763 who did not use EMM at baseline.

^EMM use at baseline (yes) was the exposure of interest.

\*\*adjusted for occupation, TB classification (bacteriologically confirmed pulmonary TB / clinically diagnosed pulmonary TB / pleurisy) and TB category (new / previously treated) using modified Poisson regression with robust variance estimates.

SAT alone, but only age, sex and health facility were adjusted for while deriving the adjusted measures of association [12]. Another study in South Africa, using SIMpill system (real time data communication with a web-based application by SMS every time the patient opens the bottle) also found that TB cure rates improved compared with control group, but the study did not describe the management approach of control group, and only 24 people were involved in the EMM group [19]. In addition, availability of mobile broadband internet may be a limitation for use in certain resource limited settings. Absence of mobile connectivity has been reported as a barrier in India [8].

There are four reasons for our observation. First, our previous study showed that the doctors from designated hospital did not shift people to DOT in nearly four-fifth instances when objective evidence of non-adherence was available. Besides missing EMM data in the EMM-IMS were common, and the guidelines did not provide clear instructions to the doctor regarding what steps need to be taken in case of missing EMM data during a month [16]. We didn't know the exact reasons for these results, but the ineffective implementation could reduce the effect of the intervention. A study from south India even showed TB treatment outcomes conversely worsened after using 99DOTS portal (the portal monitored the free missed call by the patient after taking every dose which indicated medication compliance) because of poor implementation [8].

Second, the intention to treat analysis gives conservative estimates while at the same time provides information on what would happen in a real life scenario. A significant number of people (20.6%) that used EMM at baseline stopped using EMM later during the treatment and many started using EMM after the first month (35% of those who did not use EMM from baseline).

Third, the piloting of EMM in programme settings in 30 counties of China was one of the many interventions of the comprehensive TB control model, designed by National Health Commission of the PRC—Bill & Melinda Gates Foundation Tuberculosis Prevention and Control Project (China-Gates Foundation TB Project). Other interventions that were applicable to all people with TB irrespective of EMM use may also impact the TB treatment outcomes: strengthening collaboration between centers for disease control and prevention and designated hospitals, establish new financing model for reduction of the medical burden for people with TB, integration of information systems and provision of online training for health care providers.

Finally, statistically, the baseline percentage of unfavourable outcomes and deaths were very low. Therefore, the sample size was powered enough only to detect a minimum 50% relative reduction in unfavourable outcomes (assuming baseline of 6.7%) and a minimum 67% relative reduction in deaths (assuming baseline of 3.5%). Future studies may consider assessing the effect among people with previously treated TB and other sub-groups that have a higher baseline percentage of unfavourable outcomes in China (17% with previously treated TB for 2017 cohort).

There is a need for further implementation research (including systematic qualitative enquiry) to optimize the utilization of the EMM. This may include steps to reducing missing adherence data during monthly visit to TB designated hospital or assuming non-adherence and shifting patient to DOT in the event of missing adherence data. The programme should ensure continuous use of EMM by reducing the number of people stopping EMM midway. The programme should also sensitize the doctors to take action based on EMM data which includes shifting people to DOT administered by village doctor as and when indicated [16]. Effect on TB treatment outcomes needs to be reassessed in the future after implementation of these recommendations.

Our limitation was that we relied on secondary programme data. Recording errors and residual confounding cannot be ruled out. Severity of disease and co-morbidities like diabetes, smoking and tobacco use at baseline were not adjusted for.

## Conclusion

There is limited information on the effect of EMM on TB treatment outcomes. We therefore assessed an indigenously developed EMM in China that locally recorded (within the device) information every time the box was opened to take the drugs, providing a surrogate indicator for medication adherence (no real time data). Despite, improving adherence in trial conditions, under programmatic settings, the EMM did not result in improved TB treatment outcomes when compared to SAT alone. Actions are required for better optimization by taking action on shift to DOT when indicated, addressing the issue of missing data and ensuring continuous use once started on medication monitor.

## Supporting information

**S1 Annex. Data set including the codebook.**
(XLSX)

## Acknowledgments

This research was conducted through the Structured Operational Research and Training Initiative (SORT IT), a global partnership led by the Special Program for Research and Training in Tropical Diseases at the World Health Organization (WHO/TDR). The model is based on a course developed jointly by the International Union Against Tuberculosis and Lung Disease (The Union) and Medécins sans Frontières (MSF/Doctors Without Borders). The specific SORT IT program which resulted in this publication was jointly developed and implemented by: The Union South-East Asia Office, New Delhi, India; the Centre for Operational Research, The Union, Paris, France; The Union, Mandalay, Myanmar; The Union, Harare, Zimbabwe; MSF Luxembourg Operational Research (LuxOR); MSF Operational Center Brussels (MSF OCB); Jawaharlal Institute of Postgraduate Medical Education and Research (JIPMER), Puducherry, India; Post Graduate Institute of Medical Education and Research (PGIMER), Chandigarh, India; All India Institute of Medical Sciences (AIIMS), New Delhi, India; ICMR-National institute of Epidemiology, Chennai, India; Society for Education Welfare and Action (SEWA)—Rural, Jhagadia, India; Common Management Unit (AIDS, TB & Malaria), Ministry of National Health Services, Regulations and Coordination, Islamabad, Pakistan; and Kidu Mobile Medical Unit, His Majesty's People's Project and Jigme Dorji Wangchuck National Referral Hospital, Thimphu, Bhutan. We also gratefully acknowledge all the participants who took part in this study.

## Author Contributions

**Conceptualization:** Ni Wang, Hemant Deepak Shewade, Pruthu Thekkur, Hui Zhang, Xiaolin Wang, Fei Huang.

**Data curation:** Ni Wang, Yanli Yuan, Xiaomeng Wang, Miaomiao Sun.

**Formal analysis:** Ni Wang, Hemant Deepak Shewade, Pruthu Thekkur, Miaomiao Sun, Fei Huang.

**Methodology:** Ni Wang, Hemant Deepak Shewade, Pruthu Thekkur.

**Project administration:** Ni Wang.

**Writing – original draft:** Ni Wang, Hemant Deepak Shewade, Pruthu Thekkur.

**Writing – review & editing:** Ni Wang, Hui Zhang, Yanli Yuan, Xiaomeng Wang, Xiaolin Wang, Fei Huang.

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
