## [Decision Letter · Decision Letter 0]

27 Aug 2020

PONE-D-20-13035

Whether electronic medication monitors improve tuberculosis outcomes? Programmatic experience from China

PLOS ONE

Dear Dr. Huang,

Thank you for submitting your manuscript to PLOS ONE, and many thanks for your understanding with the delay experienced in securing reviewers for your paper.

After careful consideration, we feel that it has merit but does not fully meet PLOS ONE’s publication criteria as it currently stands. Therefore, we invite you to submit a revised version of the manuscript that addresses the points raised during the review process.

In particular, the authors were interested in some methodological aspects, including an analysis of the results by stratified levels (EMM duration use, etc), some clarification about the population eligible to enrol in the study, some crucial queries regarding the ethical approval obtained for this study, and the justification of statistical significance with consistency. Additionally, I agree with the reviewer that a more robust framing of this study in the existing literature should be carried out in the Introduction, as well as the Discussion. Finally, please consider the formatting issues about references

We look forward to receiving your revised manuscript.

Kind regards,

Enrique Castro-Sánchez

Academic Editor

PLOS ONE

Journal Requirements:

Reviewers' comments:

Reviewer's Responses to Questions

**Comments to the Author**

1. Is the manuscript technically sound, and do the data support the conclusions?

Reviewer #1: Yes

Reviewer #2: Yes

2. Has the statistical analysis been performed appropriately and rigorously? 

Reviewer #1: No

Reviewer #2: Yes

3. Have the authors made all data underlying the findings in their manuscript fully available?

Reviewer #1: Yes

Reviewer #2: Yes

4. Is the manuscript presented in an intelligible fashion and written in standard English?

Reviewer #1: Yes

Reviewer #2: Yes

5. Review Comments to the Author

Reviewer #1: The study “Whether electronic medication monitors improve tuberculosis outcomes? Programmatic experience from China” is dealt with TB adherence technology and traced its comparative unfavorable outcomes with self-administered therapy. It is a retrospective cohort in design and reflects a relevant subject. However, I have some major concerns about the methodology and how the paper is written.

Title:

1. Is a bit peculiar; good to revise. And revise the abstract once you have gone through the comments in the main text.

Introduction

2. This section seems a skewed one as it relies on limited literature including a copy-past of single evidence (paragraph 2, ref 3). Authors need to consider more literature for a reasonable judgment of existing evidence.

3. It helps if the authors highlight outcomes of previous SAT studies in India and nationally approved adherence modalities. And I suggest if some descriptions under “study setting” are pooled here, mainly in paragraph 6, for a strong justification of the need of this study.

Methods

4. The authors should rewrite this section with clear sub-headings for inclusive information and potential replications elsewhere.

5. Study setting: could benefit from the 1st paragraph in “study design and population” (provinces, counties …) and this will simplify understanding.

6. Study design and population. Line 116 says “This was a cohort study design using secondary data…”. Is this to mean a retrospective cohort study?

7. Population/eligibility should be well defined.

8. Ethics. Does it mean China CDC and The Union ethics committees gave the use of secondary data a waiver of informed consent or they approved the primary study? This should not contradict with the data collection date to ensure credibility. Page 6 line 119, study design and population section, states that all the eligible people with TB during July-December 2018 were included. Additionally, the authors should include some brief information on consent to participate.

9. Tables 1 & 2 are not in their places.

10. Data analysis: Not clear why the authors preferred to use two p values (p<.05 and <.2)

Result

11. It is not clear how level of significance is interpreted across.

12. Authors need to check consistency of data in the text descriptions and tables.

Discussion

13. Comparison of findings with more literature would benefit the study.

Conclusion

14. Recheck the conclusion following reviews of results

Acknowledgment

15. This is a very exaggerated one.

References

16. This needs revision across for consistency. Some are with full names of authors, some lack the journal’s name and the like.

Reviewer #2: General comments:

This publication seeks to assess the relative differences in unfavourable outcomes and deaths among those started on EMM at baseline (within first month of diagnosis) when compared to SAT alone. The paper was well written, and the analysis well conducted. Some additions/clarification would improve the analysis interpretation.

However, the paper in its current form lacks further stratifications beyond EMM use at baseline, to fully capture the outcomes. Authors have restricted themselves to EMM use at baseline which does not tell the full story of the EMM group during treatment. The background and methods are well written, and the paper would be more informative should the results and discussion include further stratifications, at the very least stratification by duration of EMM use or stratification by those who Shift to DOT, this is available from a prior publication by the same authors (https://doi.org/10.1371/journal.pone.0232337.g003) (this is the same cohort).

General recommendation:

Revise/resubmit with revisions (especially the results)

Specific comments:

Title:

The wording of the title should be changed to frame it as a question if that is goal of the authors, otherwise it should not end with a question mark.

Background:

Line94: add (s) to solution “most accessible and affordable solution”

Line 95: rephrase “… studies have also shown a high level of acceptability…..among people”

Methods:

Line 120: replace “during” with between

Results:

The authors did not address (1) duration or (2) shift to DOT for the EMM group in a stratified analysis. This is key in assessing treatments outcomes when digital technologies are employed to enhance adherence. The authors could expand Table 4 and 5 with these results and variables “DuraEMM2” and “DuraEMM3” from the previous publication could serve as duration and “shifdot” would be used for shift to DOT. Is it possible to perform these analyses (i.e. stratified analysis by duration and/or shift to DOT)? (I understand that due to limited data this might not be possible but where necessary, please address as a limitation the small numbers in some strata. At the very least, stratification warrants a paragraph or two in the discussion because it is key to interpreting the outcomes.

Could you also look at favourable outcomes in table 5? These would further shed light on the impact of EMM during TB treatment.

Line 209: you can remove “as”

Discussion:

Most of the feedback is detailed in the results section above. You can also add more pill boxes references in the discussion as there are various studies looking at the use of these adherence technologies.

Line 257: correct setting(s) in “use in certain resource limited setting.”

Conclusion:

Line 310: correct “providing an surrogate”

6. PLOS authors have the option to publish the peer review history of their article (what does this mean?). If published, this will include your full peer review and any attached files.

Reviewer #1: **Yes: **Dr. Tsegahun Manyazewal

Reviewer #2: **Yes: **Ntwali Placide

---

## [Author Response · Author response to Decision Letter 0]

25 Sep 2020

Reviewer #1: 

The study “Whether electronic medication monitors improve tuberculosis outcomes? Programmatic experience from China” is dealt with TB adherence technology and traced its comparative unfavorable outcomes with self-administered therapy. It is a retrospective cohort in design and reflects a relevant subject. However, I have some major concerns about the methodology and how the paper is written.

REVIEWER COMMENTS 

1. Title

Is a bit peculiar; good to revise. And revise the abstract once you have gone through the comments in the main text.

AUTHOR RESPONSE

Thank you very much for this comment. We have revised the title, please see lines 1-2 of revised manuscript with track changes. Currently the revised title reads as “Do electronic medication monitors improve tuberculosis treatment outcomes? Programmatic experience from China”.

REVIEWER COMMENTS 

2. Introduction

This section seems a skewed one as it relies on limited literature including a copy-past of single evidence (paragraph 2, ref 3). Authors need to consider more literature for a reasonable judgment of existing evidence.

AUTHOR RESPONSE

Thank you. We agree with your suggestion, we have further searched and supplemented with relative literature. Besides, we have carefully gone through the manuscript and revised the introduction as you suggested. (please see lines 75-136 of revised manuscript with track changes)

REVIEWER COMMENTS 

3. Introduction

It helps if the authors highlight outcomes of previous SAT studies in India and nationally approved adherence modalities. And I suggest if some descriptions under “study setting” are pooled here, mainly in paragraph 6, for a strong justification of the need of this study.

AUTHOR RESPONSE

Thanks for the suggestion. We have revised introduction section based on your inputs. Please refer to line 85-103 of revised manuscript with track changes.

REVIEWER COMMENTS 

4. Methods

The authors should rewrite this section with clear sub-headings for inclusive information and potential replications elsewhere.

AUTHOR RESPONSE

Thank you. We have rewritten this section and let it more clear under the sub-headings (please see lines 139-187 of revised manuscript with track changes)

REVIEWER COMMENTS 

5. Methods

Study setting: could benefit from the 1st paragraph in “study design and population” (provinces, counties …) and this will simplify understanding.

AUTHOR RESPONSE

Thank you. We have revised this sentence. (please see lines 144-149 of revised manuscript with track changes)

REVIEWER COMMENTS 

6. Methods

Study design and population. Line 116 says “This was a cohort study design using secondary data…”. Is this to mean a retrospective cohort study?

AUTHOR RESPONSE

Thank you. This is a retrospective cohort study design using secondary data. EMM supported SAT was offered to all the eligible people at the beginning of outpatient treatment under the consent of patients. We extracted data variables from TBIMS and EMM-IMS, which were regularly online, reported and quality controlled. However, the STROBE guidelines do not recommend the use of retrospective or prospective in the study design. Hence, we have not used the word ‘retrospective’ in the study design. We hope this is fine. 

REVIEWER COMMENTS 

7. Methods

Population/eligibility should be well defined.

AUTHOR RESPONSE

Thank you. We have revised this sentence, clearly defined the ‘eligible’ people. (please see lines 175-177 of revised manuscript with track changes)

REVIEWER COMMENTS 

8. Methods

Ethics. Does it mean China CDC and The Union ethics committees gave the use of secondary data a waiver of informed consent or they approved the primary study? This should not contradict with the data collection date to ensure credibility. Page 6 line 119, study design and population section, states that all the eligible people with TB during July-December 2018 were included. Additionally, the authors should include some brief information on consent to participate.

AUTHOR RESPONSE

Thank you very much for this comment. Yes, China CDC and The Union ethics committees approved this primary study and the data collected from routine online system (secondary data). As the implementation was done in routine programmatic settings, the experience which we are sharing in this paper and the data used involved secondary data routinely captured in the programme, both the ethics committees waived the need for written informed consent. We have included information on consent to participate. We hope this is fine. (please see lines 211-212 of revised manuscript with track changes) 

REVIEWER COMMENTS 

9. Methods

Tables 1 & 2 are not in their places.

AUTHOR RESPONSE

Thank you. We have revised it. (please see lines 220-223 and 236-242 of revised manuscript with track changes)

REVIEWER COMMENTS 

10. Methods

Data analysis: Not clear why the authors preferred to use two p values (p<.05 and <.2)

AUTHOR RESPONSE

We have revised this. To determine potential confounders, both p value cut offs have been used as p<0.2. In addition age and sex were included as universal confounders. There has been no change in the interpretation of results as a result of this minor change. (please see lines 251 of revised manuscript with track changes)

REVIEWER COMMENTS 

11. Result

It is not clear how level of significance is interpreted across.

AUTHOR RESPONSE

If 95% CIs cross each other (for two proportions) or the 95% CIs cross the null value (RR=1), then the results are considered to be statistically insignificant (this corresponds to p>/=0.05). 

If 95% CIs do not cross each other (for two proportions) or the 95% CIs do not cross the null value (RR=1), then the results are considered to be statistically significant (this corresponds to p<0.05). 

We hope this is fine. 

REVIEWER COMMENTS 

12. Result

Authors need to check consistency of data in the text descriptions and tables.

AUTHOR RESPONSE

Thank you. We have checked this and made corrections. (please see lines 318-327 of revised manuscript with track changes)

REVIEWER COMMENTS 

13. Discussion

Comparison of findings with more literature would benefit the study.

AUTHOR RESPONSE

Thank you. We have added some more references and revised the discussion part. Please refer to line 334-356 of revised manuscript with track changes. 

REVIEWER COMMENTS 

14. Conclusion

Recheck the conclusion following reviews of results.

AUTHOR RESPONSE

Thank you for the comment. There has been no change in results or its interpretation after the review of results.

REVIEWER COMMENTS 

15. Acknowledgment

This is a very exaggerated one.

AUTHOR RESPONSE

This study was supported by SORT IT, a training model organized by The Union and other stakeholders. SORT IT is an operational research training initiative endorsed by WHO and this is a standard requirement. We hope this is fine.

REVIEWER COMMENTS 

16. References

This needs revision across for consistency. Some are with full names of authors, some lack the journal’s name and the like.

AUTHOR RESPONSE

Thank you very much for pointing this out. Our sincere apologies for this. We have revised this part. (please see lines 473-526 of revised manuscript with track changes)

Reviewer #2: 

REVIEWER COMMENTS

1. General comments

This publication seeks to assess the relative differences in unfavourable outcomes and deaths among those started on EMM at baseline (within first month of diagnosis) when compared to SAT alone. The paper was well written, and the analysis well conducted. Some additions/clarification would improve the analysis interpretation.

However, the paper in its current form lacks further stratifications beyond EMM use at baseline, to fully capture the outcomes. Authors have restricted themselves to EMM use at baseline which does not tell the full story of the EMM group during treatment. The background and methods are well written, and the paper would be more informative should the results and discussion include further stratifications, at the very least stratification by duration of EMM use or stratification by those who Shift to DOT, this is available from a prior publication by the same authors (https://doi.org/10.1371/journal.pone.0232337.g003) (this is the same cohort).AUTHOR RESPONSE

Thank you for the comment. We discussed this stratification issues among the authors and decided to restrict our analysis into two groups. We think under programmatic setting, intention to treat analysis is the most feasible, so we stratified those eligible people into two groups: i) started EMM in first months ii) not started EMM in first month. We decided against dividing those who were eligible for EMM by‘duration’or by ‘ever EMM use’ because that will induce a selection bias, those who partially used EMM (involved starting EMM after first month) will have better outcomes than those who used EMM starting from baseline. We have clarified this further in lines xxx of revised manuscript with track changes. The description of EMM use, missing data, shift to DOT based on EMM data has been mentioned in detail in our previous paper (mentioned by you). We have also summarized the findings of this previous paper in the lines 107-111 (of the introduction) in the revised manuscript with track changes. 

REVIEWER COMMENTS 

2. Title

The wording of the title should be changed to frame it as a question if that is goal of the authors, otherwise it should not end with a question mark.

AUTHOR RESPONSE

Thanks for the suggestion. We have revised it. Please refer to lines 1-2 of revised manuscript with track changes. The revised title is as follows

Do electronic medication monitors improve tuberculosis treatment outcomes? Programmatic experience from China.

REVIEWER COMMENTS 

3. Background

Line94: add (s) to solution “most accessible and affordable solution”

AUTHOR RESPONSE

Thank you. We have revised it. Please refer to lines 89 of revised manuscript with track changes. 

REVIEWER COMMENTS 

4. Background 

Line 95: rephrase “… studies have also shown a high level of acceptability…..among people”

AUTHOR RESPONSE

Thank you. We have revised it. Please refer to lines 106 of revised manuscript with track changes. 

REVIEWER COMMENTS 

5. Methods

Line 120: replace “during” with between 

AUTHOR RESPONSE

Thank you. We have revised it. Please refer to lines 179 of revised manuscript with track changes. 

REVIEWER COMMENTS 

6. Results 

The authors did not address (1) duration or (2) shift to DOT for the EMM group in a stratified analysis. This is key in assessing treatments outcomes when digital technologies are employed to enhance adherence. The authors could expand Table 4 and 5 with these results and variables “DuraEMM2” and “DuraEMM3” from the previous publication could serve as duration and “shifdot” would be used for shift to DOT. Is it possible to perform these analyses (i.e. stratified analysis by duration and/or shift to DOT)? (I understand that due to limited data this might not be possible but where necessary, please address as a limitation the small numbers in some strata. At the very least, stratification warrants a paragraph or two in the discussion because it is key to interpreting the outcomes. 

AUTHOR RESPONSE

Thank you for the comment. We have stuck to the intention to treat analysis based on EMM use among eligible patients at baseline. There are multiple subgroups among whom stratified analysis may be provided and crude rates of unfavourable outcomes or deaths in these sub-group could be calculated. However these crude rates are prone to be mis-understood and mis-interpreted on their face value. For example the unfavourable outcomes among those who started EMM later (after one month) and in those who used EMM from baseline. The unfavourable outcomes in the former are lower than the latter and this is due to selection bias (survival benefit for those who start EMM after one month). Hence, in lines 160-171 of revised manuscript with track changes, we have explained why we strict to divide the patients into two groups based on baseline EMM use. We hope this is fine. 

REVIEWER COMMENTS 

7. Results

Could you also look at favourable outcomes in table 5? These would further shed light on the impact of EMM during TB treatment. 

AUTHOR RESPONSE

Thank you. The impact of EMM on TB outcomes will not change irrespective of whether we have unfavourable or favourable outcomes as our outcome of interest. If there is no association between EMM use (yes) and unfavourable outcome, that means there is no association between EMM and outcomes. This association will not change irrespective of which outcome we use as our outcome of interest. 

For example, if we take favourable outcome as the outcome of interest, then the aRR for favourable outcome is also one. If we take ‘no death’ as the outcome of interest then the aRR will be around 1.4 and this will not be statistically significant also. 

REVIEWER COMMENTS 

8. Results

Line 209: you can remove “as”

AUTHOR RESPONSE

Thank you. We have revised it. Please refer to lines 290 of revised manuscript with track changes. 

REVIEWER COMMENTS 

9. Discussion

Most of the feedback is detailed in the results section above. You can also add more pill boxes references in the discussion as there are various studies looking at the use of these adherence technologies.

AUTHOR RESPONSE

Thanks for the suggestion. We have add several references and revised the discussion part. Please refer to line 334-356 of revised manuscript with track changes.

REVIEWER COMMENTS 

10. Discussion

Line 257: correct setting(s) in “use in certain resource limited setting.”

AUTHOR RESPONSE

Thank you. We have revised it. Please refer to lines 355 of revised manuscript with track changes. 

REVIEWER COMMENTS 

11. Conclusion 

Line 310: correct “providing an surrogate”

AUTHOR RESPONSE

Thank you for the comment. We have replaced ‘an’ with ‘a’. Please refer to lines 409 of revised manuscript with track changes.

---

## [Decision Letter · Decision Letter 1]

27 Oct 2020

Do electronic medication monitors improve tuberculosis treatment outcomes? Programmatic experience from China

PONE-D-20-13035R1

Dear Dr. Huang,

We’re pleased to inform you that your manuscript has been judged scientifically suitable for publication and will be formally accepted for publication once it meets all outstanding technical requirements.

Kind regards,

Enrique Castro-Sánchez

Academic Editor

PLOS ONE

Additional Editor Comments (optional):

Reviewers' comments:

Reviewer's Responses to Questions

**Comments to the Author**

1. If the authors have adequately addressed your comments raised in a previous round of review and you feel that this manuscript is now acceptable for publication, you may indicate that here to bypass the “Comments to the Author” section, enter your conflict of interest statement in the “Confidential to Editor” section, and submit your "Accept" recommendation.

Reviewer #1: All comments have been addressed

2. Is the manuscript technically sound, and do the data support the conclusions?

Reviewer #1: Yes

3. Has the statistical analysis been performed appropriately and rigorously? 

Reviewer #1: Yes

4. Have the authors made all data underlying the findings in their manuscript fully available?

Reviewer #1: Yes

5. Is the manuscript presented in an intelligible fashion and written in standard English?

Reviewer #1: Yes

6. Review Comments to the Author

Reviewer #1: The authors have addressed my comments and revised the manuscript accordingly. I have no additional comments.

7. PLOS authors have the option to publish the peer review history of their article (what does this mean?). If published, this will include your full peer review and any attached files.

Reviewer #1: **Yes: **Dr. Tsegahun Manyazewal

---

## [Editor Report · Acceptance letter]

29 Oct 2020

PONE-D-20-13035R1 

Do electronic medication monitors improve tuberculosis treatment outcomes? Programmatic experience from China 

Dear Dr. Huang:

I'm pleased to inform you that your manuscript has been deemed suitable for publication in PLOS ONE. Congratulations! Your manuscript is now with our production department. 

Kind regards, 

on behalf of

Dr. Enrique Castro-Sánchez 

Academic Editor

PLOS ONE